# 🚌 BusMA: A Bus Communication Substrate for Multi-Agent Systems

## Abstract

Multi-Agent (MA) systems are effective at solving complex tasks that demand advanced planning, tool use, and the synthesis of evidence from multiple sources. Existing systems typically use Hierarchical Manager-Worker (HMW) structures or Router-based Message Passing (RMP) as their communication protocol to coordinate work. However, they can be restricted due to communication ineffectiveness, since agents cannot directly consult specific colleagues or operate beyond assigned subtasks, and misrouted messages can propagate errors. Inspired by bus communication systems in computer systems, we propose BusMA, a bus communication MA framework that allows any agent to address specific peers. Our framework comprises a *Chair* agent, *Worker* agents, and a *communication bus*. *Worker* agents perform multi-step reason–act–call interactions, enabling targeted requests for help or critique, with the *Chair* agent synthesizing insights from all agents' communications while adding its own reasoning to produce coherent solutions. The *communication bus* routes addressable messages and executes requests. Across two frontier LLMs and benchmarks spanning diverse domains, including image understanding, mathematics, and knowledge-based tasks, as well as GAIA with tasks of varied complexity, BusMA consistently achieves the best results, outperforming state-of-the-art multi-agent communication approaches (HMW and RMP-based methods). Anonymous code is available at https://anonymous.4open.science/r/Bus-MA-370E.

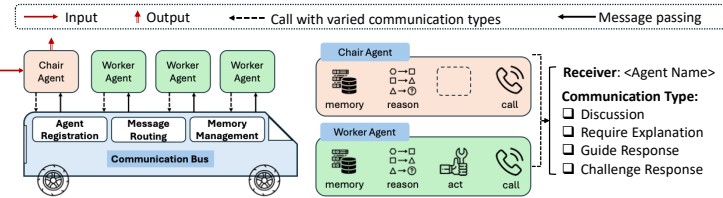

Figure 1: Overview of the BusMA communication substrate.

## 1 Introduction

Multi-Agent (MA) systems, comprising a set of agents that can autonomously reason, act and communicate, powered by large language models (LLMs) (Achiam et al., 2023; Comanici et al., 2025; DeepSeek-AI et al., 2025), are effective in solving complex real-world tasks. These tasks, such as mathematical reasoning (Lei et al., 2024) and reasoning-grounded information retrieval (Huang et al., 2025), often require planning, tool use (e.g., web search API) and synthesis of evidence from diverse resources (Fang et al., 2025; Du et al., 2025). MA systems rely on specialized subagents to address these complex tasks in a collaborative manner. Hence, their effectiveness depends on the communication quality between agents and their coordination strategy (Guo et al., 2024; Chen et al., 2023; Liang et al., 2023; Du et al., 2023; Wu et al., 2023b).

Existing MA communication protocols can be broadly categorized into *Hierarchical Manager-Worker (HMW) structure* and *Router-based Message Passing (RMP)*. HMW employs a hierarchical control structure, where a designated manager agent addresses the decomposition of complex tasks and assigns sub-tasks to subordinate agents with specific capabilities, e.g., code execution or web search (Qian et al., 2024; Gu et al., 2025; Zhang et al., 2025). While effective for simple tasks involving, this paradigm limits agent communication to one-way task allocation and result submission, thereby preventing richer peer dialogue. In contrast, RMP introduces a central router agent, which mediates all communication. Agents send their messages to the router, which then forwards to selected recipients according to predefined rules. The router often maintains a shared memory of past interactions, allowing it to route messages consistently and provide agents with an overview of global states (Wu et al., 2023a; Nonomura & Mori, 2025). This approach enables flexible workflows, yet a shared memory pool is forced through the central router, which prevents native peer dialogue. The centralized nature of HMW and RMP gives rise to two primary limitations. First, communication is constrained: agents cannot directly consult specific colleagues or act autonomously beyond their assigned subtasks, preventing the exchange of explanations, critiques, and joint problem-solving, which are essential for effective teamwork (Ke et al., 2025; Jin et al., 2025; Krishnan, 2025). Second, these designs are prone to error propagation. The router may misroute messages to wrong or sub-optimal peers, and the reliance on long context to maintain shared memory can propagate errors (Piatti et al., 2024; Han et al., 2024; Maragheh & Deldjoo, 2025; Sagirova et al., 2025).

To address these limitations, we propose BusMA, a bus communication substrate for MA systems. The bus architecture in computer systems (Patterson & Hennessy, 2017) enables components to communicate directly through a shared communication channel, eliminating the need for point-to-point connections. This architecture removes centralized controllers and allows components to autonomously initiate data transfers based on their needs. In MA systems, such an architecture translates to enabling agents to directly address specific peers through a communication bus, thereby eliminating centralized bottlenecks from managers and routers. This approach allows agents to autonomously initiate information exchange beyond assigned tasks and establishes a standard protocol for cross-framework interoperability. BusMA consists of three core components: *Worker agents*, a specialized *Chair agent*, and the *communication bus*. A worker agent, equipped with access to a "personal" memory, first produces a *low-level insight*, which is a reasoning outcome derived from its own memory. Subsequently, it can choose to issue a call directed to a different peer with one or more of four communication types (i.e., discussion, explanation, challenge, and guidance). The response to this interaction yields a *mid-level insight*, which reflects reasoning enriched through peer input and is recorded to the bus. The Chair agent is a specialized Worker agent with acting disabled. Its main responsibility is to integrate messages from other agents with its own reasoning to synthesize insights and steer task progress. Specifically, it fuses the bus history with its memory to form *high-level insights* that guide coordination and final answer generation. The communication bus routes addressable messages and maintains a centralized memory managed by the chair. Figure 1 shows the overall BusMA framework.

**Contributions.** BusMA is a novel bus communication substrate that addresses existing system limitations through two key mechanisms: (1) extended communication types that enable varied interaction modes beyond simple task assignment; (2) multi-level insight generation across individual (low), dialogue (mid), and synthesis (high) levels. Through comprehensive experiments across 12 datasets spanning image analysis, mathematics, and knowledge question answering, we show that BusMA outperforms state-of-the-art methods. Additional experiments show that it consistently outperforms previous methods on general agentic tasks.

## 2 RELATED WORK

**LLM Based Agents.** LLMs such as GPT (Achiam et al., 2023), Gemini (Comanici et al., 2025), and DeepSeek (DeepSeek-AI et al., 2025), serve as the foundational architecture for autonomous agent development. Agentic architectures augment the base LLM with complementary mechanisms, including advanced planning strategies for task decomposition (Wei et al., 2022; Huang et al., 2024; Li et al., 2025; Hu et al., 2025; Erdogan et al., 2025), external tool use and knowledge bases (Zhang et al., 2024b; Wu et al., 2024a; Qin et al., 2024; Feng et al., 2025), and long-term memory or reflection mechanisms for persistent state and iterative improvement (Shinn et al., 2023; Modarressi et al., 2023; Zhong et al., 2024; Mei et al., 2024; Xu et al., 2025; Chhikara et al., 2025). This architectural

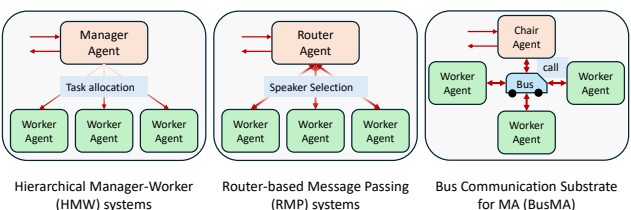

Figure 2: Comparison between different MA frameworks.

framework has been applied to diverse domains such as automated programming (Trivedi et al., 2024; Zhang et al., 2024a; Chen et al., 2025), system interaction (Wu et al., 2024b; Bonatti et al., 2024), and scientific discovery (Hong et al., 2024; Novikov et al., 2025). However, single-agent LLMs remain brittle on long-horizon, interdependent tasks, with evaluations reporting systematic failures in planning, decision-making, and instruction following (Liu et al., 2024; Xie et al., 2024; Wang et al., 2025). Consequently, MA systems have emerged as a promising approach to improve reasoning accuracy and task completion by distributing complex problems among specialized agents (Li et al., 2023; Du et al., 2024).

**LLM Based Multi Agent Systems.** Building on advances in single-agent development, researchers have explored MA systems composed of interacting LLM agents. Early studies highlighted the value of agent dialogue for social simulation and collaborative problem-solving (Park et al., 2023; Li et al., 2023), leading to the development of more structured communication protocols. These systems generally fall into two categories: *Hierarchical Manager-Worker (HMW) Systems* and *Router-based Message Passing (RMP) Systems*. HMW employs a hierarchical structure with a central manager agent that decomposes tasks and assigns subtasks to subordinate worker agents (Chen et al., 2023; Qian et al., 2024). Examples include systems where a planner orchestrates tool calls for an executor (Lu et al., 2025), or where a manager invokes agents as callable tools or workers for individual task steps (Roucher et al., 2025; LangChain Inc., 2025). RMP by contrast, such as AutoGen (Wu et al., 2023a), uses a central router agent, often an LLM, to dynamically select the next agent to continue the task (a process named as speaker selection) without explicit task decomposition, while maintaining a shared memory pool for coordination.

While effective, centralized MA designs confine communication to task assignment and reporting channels, limiting the richer peer-to-peer exchanges. The reliance on single control points also introduces bottlenecks and propagates errors across the dialogue, while dependence on shared memory can further amplify inaccuracies. These limitations highlight the need for more flexible and addressable communication strategies, which motivates our introduction of BusMA, a bus communication substrate that supports targeted peer consultation and coherent MA reasoning. Figure 2 comparatively illustrates existing paradigms and BusMA for MA communication.

## 3 THE BusMA COMMUNICATION FRAMEWORK

In computer systems, bus architectures offer a flexible way to connect multiple components: each component communicates through a shared bus using a standardized protocol, without relying on a central controller to manage every interaction. This design enables components to exchange information directly and flexibly, while still preserving overall system coherence (Patterson & Hennessy, 2017). Inspired by this architectural principle, we propose BusMA, a bus-style communication substrate for MA systems. By enabling agents to engage in flexible and interactive dialogue, requesting clarifications, challenging assumptions, and providing expertise-based guidance, BusMA aims to transform MA collaboration from rigid task distribution into dynamic knowledge synthesis through diverse conversation. This approach enables agents to iteratively refine reasoning and adapt solutions to complex task requirements in real time.

The BusMA framework consists of a communication bus and two main agent types: (1) worker agents, responsible for collaboration and task execution; and (2) a chair agent, a specialized worker agent for information synthesis. In the following subsections, we first introduce the design of worker and chair agents, and then describe the communication bus that coordinates their interactions.

## 3.1 Worker Agent

Worker agents are the core operational units of BusMA. They are designed to reason, act, and communicate with peers over the bus by sending and receiving messages, forming the basis of collaborative problem solving. Each agent $\alpha_i$ is defined as $\alpha_i = (\mathcal{L}, \mathcal{P}_i, \mathcal{M}_i, \mathcal{K}_i(t), \mathcal{T}_i, \mathcal{N}, T_{max})$ where $\mathcal{L}$ is an LLM, $\mathcal{P}_i$ is the base prompt that defines the agent's role, output format, and specifically designates the communication types that can be registered to the bus. $\mathcal{M}_i$ represents messages received from the bus, $\mathcal{K}_i(t)$ is the agent's personal memory at iteration $t$, $\mathcal{T}_i$ lists available tools, $\mathcal{N}$ indicates a list of accessible peer agents, and $T_{max}$ specifies the maximum number of iterations.

**Activation and Initialization.** When a worker receives a message from the bus, it activates and begins its execution cycle ($t = 0$). It constructs its initial prompt by concatenating the base prompt $\mathcal{P}_i$ with the received message $\mathcal{M}_i$, as well as the descriptions of the peer agent list $\mathcal{N}$ and available tools $\mathcal{T}_i$. By modifying the base prompt $\mathcal{P}_i$ and tool list $\mathcal{T}_i$, our BusMA allows the instantiation of workers with diverse capabilities tailored to specific tasks (e.g., search, code execution). Upon initialization, each worker first *reasons* and then can *act*, following the ReAct framework (Yao et al., 2023), or *call*.

**Reason.** The LLM backbone $\mathcal{L}$ analyzes the current context to produce a low-level insight $r_t^{\text{low}}$ at cycle $t$. This captures the agent's local understanding of the task and informs the choice of action based on the current context and available resources. The agent then selects one of two actions: Act and Call. The decision between the two actions emerges from the agent's reasoning about the current context: Act is selected when the interaction with available tools is required to gather information or perform operations, whereas Call is selected when communication with another agent is needed for discussion, critique, requesting clarification, or seeking guidance.

**Act.** $\mathcal{L}$ identifies a tool $\tau_j \in \mathcal{T}_i$ and autonomously configures its execution arguments $\theta$ (e.g., file path or a formulated search query) based on the current context and task requirements. Then, the agent executes the tool and receives an observation $o_t = \tau_j(\theta)$. For instance, if tasked with finding recent research papers on a specific topic, the agent may use a search tool with a query as $\theta$, obtaining a list of publications as the observation. The agent then updates its memory with both the reasoning insights and the observation: $\mathcal{K}_i(t+1) = \mathcal{K}_i(t) \cup \{r_t^{\text{low}}, o_t\}$. Next, it proceeds to iteration $t + 1$, reconstructing its prompt with $\mathcal{P}_i$, $\mathcal{M}_i$, $\mathcal{N}$, $\mathcal{T}_i$, and the updated memory $\mathcal{K}_i(t+1)$. $\mathcal{L}$ analyzes this accumulated information, producing a new low-level insight $r_{t+1}^{\text{low}}$, and selects the next action. This process continues, with the agent alternating between reasoning and acting while accumulating observations and insights in its memory. The cycle terminates under two conditions: (1) if the agent chooses Call, or (2) the agent reaches the maximum iteration limit $T_{max}$, in which case the agent returns the error message "no message provided" to the agent that originally sent it the message.

**Call.** $\mathcal{L}$ selects a target agent $\alpha_j \in \mathcal{N}$ to send a message. The agent generates an action $a_t^{\text{call}} = (\alpha_j, m_{i \to j})$, where $\alpha_j$ is the receiver and $m_{i \to j}$ is the composed message formatted in JSON which is sent to the bus. Subsequently, the agent's current execution cycle terminates. The Call action considers four communication types, determined by the agent's reasoning: **discussion** for bidirectional information exchange, **request for explanation** when clarifying ambiguous context, **challenge** to verify questionable results, and **guidance** when requiring help. For instance, worker agents may initiate a discussion to verify facts or issue a challenge when a peer agent's message conflicts with their own analysis. Call actions lead to mid-level insights $r_t^{\text{mid}}$ that the agent derives from its accumulated low-level insights. While low-level insights $r_t^{\text{low}}$ capture the agent's reasoning at each iteration step, mid-level insights emerge when the agent synthesizes these local observations to determine both the necessity and the form of inter-agent communication. Appendix D.1 presents an example prompt for the worker agent.

## 3.2 Chair Agent

The chair agent is a specialized worker agent that serves as the coordinator and entry point of the system. Unlike worker agents, it does not use tools (i.e., $\mathcal{T}_{\text{chair}} = \emptyset$, see Figure 1), focusing on managing collaboration. Unlike existing manager- and router-based systems, the chair does not offer central planning, or decompose and assign subtasks. It operates in two distinct modes with different

prompts: $\mathcal{P}_{\text{chair}}^{\text{COOR}}$ for the coordination (COOR) phase and $\mathcal{P}_{\text{chair}}^{\text{SUBM}}$ for the submission (SUBM) phase. Appendix D.2 presents the two-phase prompts for the Chair agent.

**Coordination Mode.** At $t = 0$, the chair agent receives a task and begins the coordination phase. It constructs its prompt by combining $\mathcal{P}_{\text{chair}}^{\text{COOR}}$ with the task description and the description of available agents $\mathcal{N}$ in a list. At each iteration $t$, the chair chooses between reason and call actions without the option of acting:

- **Reason.** If $t = 0$, the chair agent solely analyzes the available information (e.g., task description and $\mathcal{P}_{\text{chair}}^{\text{COOR}}$) to draw high-level insights. If $t > 0$, the agent has been called by other worker agents. In that case, it considers all historical messages $\mathcal{H}(< t)$ accumulated from the bus, which contain the workers' responses and their inter-agent communications, synthesizes the mid-level insights to produce high-level insights $r_{chair,t}^{\text{high}}$. Instead, if Reason happens in the initial stage ($t = 0$), the chair solely draws high-level insights.

- **Call.** The chair outputs a JSON structure specifying receiver $\alpha_j$ and message $m_{\text{chair} \rightarrow j}$, and sends this to the bus. The chair enters a listening mode to wait for future requests. At this stage, worker agents may communicate with each other, using Call actions exchanging mid-level insights $r_{i \rightarrow j,t}^{\text{mid}}$ derived from their communication and accumulated low-level insights. The chair agent is reactivated when a worker agent sends it a message. Upon reactivation, the chair agent goes in Coordination mode.

The chair can call workers using the same four communication types: **discussion**, **request for explanation**, **challenge**, and **guidance**. This protocol creates a three-tier hierarchy of insights. Low-level insights $r_t^{\text{low}}$ capture individual agents' reasoning about tool use and immediate context. Mid-level insights $r_{i \rightarrow j,t}^{\text{mid}}$ emerge from worker agents' decisions to communicate, synthesized from their accumulated low-level insights. High-level insights $r_{\text{chair},t}^{\text{high}}$ represent the chair agent's synthesis of worker communications into a global task progress understanding (Figure 3). This iterative process continues until either the chair agent determines the task is complete and generates a JSON output with a "submit" field to signal readiness for final answer generation, or the maximum iteration limit $T_{max}$ is reached. Both conditions trigger the transition to the submission phase.

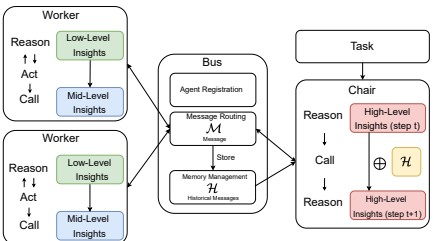

Figure 3: The formation process of the three-layer insights.

**Submission Mode.** In the submission mode, the chair agent uses the prompt $\mathcal{P}_{\text{chair}}^{(SUBM)}$ concatenated with the original task description and all accumulated insights from the coordination phase to generate the finalized solution as output. This is a single-step, irreversible action where the chair agent must produce the final answer based on all gathered information.

## 3.3 COMMUNICATION BUS

The bus provides a shared substrate for agent interaction consisting of *agent registration*, *message routing*, and *memory management* modules.

**Agent Registration Module.** When the system initializes, the agent registration module registers each instantiated worker agent and chair agent $\alpha_i$ to the bus and assigns the agent's name as its address $\mathcal{A}_i$. This creates an address registry $\mathcal{R} = \{(\alpha_i, \mathcal{A}_i) | \alpha_i \in \mathcal{N}\}$, enabling the agents to be accessed as message receivers.

**Message Routing Module.** When an agent $\alpha_i$ chooses the call action and generates $a_t^{\text{call}} = (\alpha_j, m_{i \rightarrow j})$ to send a message to the bus, the bus parses the message and extracts the receiver name $\alpha_j$, then matches it against the address registry $\mathcal{R}$. This step verifies whether $\alpha_i$ has correctly specified the receiver agent's name. Upon successful matching, the bus augments the message to create an enriched message structure $\hat{m}_{i \rightarrow j}(t) = (m_{i \rightarrow j}, \text{ID}, t, \alpha_i)$, where ID is a unique identifier, $t$ is the timestep, and $\alpha_i$ refers to the sender agent. The bus forwards $\hat{m}_{i \rightarrow j}(t)$ to the receiver $\alpha_j$, and simultaneously transfers this augmented message to the memory management module.

**Memory Management Module.** This module maintains a chronological list $\mathcal{H}(< t)$ that stores all messages passing through the bus until time $t$. It appends augmented messages received from the message routing module to the list sequentially, preserving the complete communication history. When the chair agent is designated as the receiver (that is, when any worker agent sends a message to the chair), the memory management module transfers the complete historical messages $\mathcal{H}(< t)$ to the chair agent. This enables the chair agent to access all accumulated communications between agents for synthesizing high-level insights during its coordination phase.

## 4 EXPERIMENTAL SETUP

### 4.1 BENCHMARKS

We evaluate BUSMA on two complementary groups of benchmarks: one assessing breadth across modalities and task types, and the other emphasizing depth through real-world problem solving.

**Diversity-oriented.** We evaluate twelve standard benchmarks similar to Lu et al. (2025), covering three categories: (1) *visual and spatial reasoning*, including AlgoPuzzleVQA (Ghosal et al., 2025), Hallusion-VD (Guan et al., 2024), PuzzleVQA (Chia et al., 2024), and VQA 2.0 (Goyal et al., 2017); (2) *mathematical reasoning and multi-step problem solving*, including Game of 24 (Nathan Lile, 2025), Omni-MATH (Gao et al., 2024), CLEVR-Math (Lindström & Abraham, 2022), and MathVista (Lu et al., 2023); and (3) *knowledge-based question answering* benchmarks with GPQA (Rein et al., 2024), MMLU-Pro (Wang et al., 2024), SciFIBench (Roberts et al., 2024), and HotpotQA (Yang et al., 2018). We randomly sample 200 instances from each dataset following Lu et al. (2025). Table 1 shows the modalities, domains, and reasoning skills required for each dataset.

| Datasets | Modality | Domain | 🖼 | 🟦 | 🔍 | 📑 |
|---|---|---|---|---|---|---|
| AlgoPuzzleVQA | Vision | General | ✓ | | | ✓ |
| Hallusion-VD | Vision | General | ✓ | | | |
| PuzzleVQA | Vision | General | ✓ | | | ✓ |
| VQA 2.0 | Vision | General | ✓ | | | ✓ |
| Game of 24 | Text | Mathematical | | ✓ | | ✓ |
| Omni-MATH | Text | Mathematical | | ✓ | ✓ | ✓ |
| CLEVR-Math | Vision | Mathematical | ✓ | ✓ | | ✓ |
| MathVista | Vision | Mathematical | ✓ | ✓ | ✓ | ✓ |
| GPQA | Text | Knowledge | | ✓ | ✓ | ✓ |
| MMLU-Pro | Text | Knowledge | | | ✓ | ✓ |
| SciFIBench | Vision | Knowledge | ✓ | | ✓ | |
| HotpotQA | Text | Knowledge | | | ✓ | ✓ |

Table 1: Modalities, domain and reasoning skills required (visual understanding 🖼, numerical calculation 🟦, knowledge retrieval 🔍, and multi-step reasoning 📑).

**Complexity-oriented.** To assess BusMA in addressing tasks in varied complexity, we use the **GAIA** (Mialon et al., 2024) benchmark, which includes real-world tasks requiring combinations of file parsing, web browsing, and code execution for complex problem-solving settings. The dataset is designed to be difficult for LLMs and aims to evaluate general AI assistants and agents. We use the complete GAIA validation set of 165 questions across all difficulty levels. We report task completion accuracy as the primary evaluation metric with additional tracking of API token cost to quantify communication overhead and computational efficiency.

### 4.2 MULTI AGENT SYSTEM CONFIGURATION

To implement the agents and integrated tools, we use publicly available foundation models: **DeepSeek-V3** (DeepSeek-AI et al., 2025) and **Gemini-2.5-Flash** (Comanici et al., 2025). For Gemini-2.5-Flash, we adopt a unified configuration across all agents and tools in both BusMA and the baselines. Since DeepSeek-V3 does not support image inputs, visual question answering is handled by the multimodal Gemini-2.0-Flash (Google DeepMind, 2024) due to API constraints, configured with the same parameters as DeepSeek-V3.

**BusMA System Configuration.** In **diversity tasks**, we instantiate three worker agents, each with a specific role (as per customized prompts and distinct tool sets): a **WebAgent**, equipped with `GoogleSearchTool` and `WikiSearchTool`; an **ImageQAAgent**, which directly queries a multimodal LLM (Gemini-2.5-Flash or Gemini-2.0-Flash for DeepSeek-V3), and a **CodeAgent**, which generates code at the `Act` stage and executes it. For the chair agent within BusMA, it operates with a maximum of 10 iteration steps (i.e., $T_{max}^{chair} = 10$). We set the maximum iteration steps $T_{max}^{worker} = 5$ for efficiency. In **GAIA** (complexity) benchmark, we use four worker agents: a **CodeAgent**; a **FileAgent** enables

document browsing and access to local files; and a **BrowserAgent** to address the online information retrieval operations. Following (Roucher et al., 2025), BrowserAgent operates with 20 iteration steps ($T_{max} = 20$), while all other agents, including the chair agent, use 12 steps ($T_{max} = 12$).

**Third-Party Integration.** To incorporate external agents, BusMA provides lightweight adapters that ensure compatibility with the bus protocol while preserving the agents' native functionality. Each adapter implements three core methods: (1) `register_agent` assigns the external agent a unique address on the bus, and records it in the registry. (2) `receive_message` listens for bus messages directed to the agent, extracts relevant content, and queues it for processing. (3) `handle_message` invokes the agent's native execution method with the extracted inputs and packages the outputs into standardized bus-compatible messages. This design decouples message translation from agent execution, requiring only lightweight wrapping rather than modifying the original agent logic. As a result, agents from external frameworks such as SmolAgents can be seamlessly integrated. In our experiments, both the FileAgent and BrowserAgent are connected via such adapters.

The detailed experimental settings are presented in Appendix B and the prompts of the agents are included in Appendix D.

### 4.3 BASELINES

**Diversity-oriented Benchmarks.** *OctoTools* (Lu et al., 2025) uses a centralized planner-executor architecture with 50 total iteration steps. *SmolAgents* (Roucher et al., 2025) and *LangGraph* (LangChain Inc., 2025) implement manager-worker hierarchies (i.e., HMW) for coordinating specialized agents. *AutoGen* (Wu et al., 2023a) employs router-based communication with shared memory pools. For iteration limits, SmolAgents matches BusMA's configuration (manager: 10 steps, workers: 5 steps each), while LangGraph and AutoGen use 50 total iteration steps as they do not support per-agent step definitions. We maintain a consistent core toolset across all frameworks to isolate communication architecture effectiveness rather than tool optimization capabilities.

**Complexity-oriented benchmark** For experiments on GAIA, we use Gemini-2.5-Flash-FunctionCalling and Gemini-2.5-Pro-FunctionCalling[1] to quantify the contribution of MA coordination versus single-model function calling, determining whether MA systems provide measurable benefits over monolithic approaches for complex tasks. We also compare against two state-of-the-art MA systems. *MagenticOne* (Fourney et al., 2024), developed from AutoGen, employs four specialized agents for orchestration, file browsing, web navigation, and code execution, and enforces a system-wide limit of 120 iteration steps. *OpenDeepResearch* (Roucher et al., 2025), developed from SmolAgents, uses a Manager agent (12 steps) that coordinates a BrowserAgent (20 steps) for complex reasoning tasks. OpenDeepResearch is a fixed framework with predefined agent components and tools that cannot be changed. The detailed baseline settings are presented in Appendix C.

## 5 RESULTS AND ANALYSIS

**Diversity Benchmarks.** Table 2 shows the results across models and tasks. We observe that BusMA obtains the highest overall average accuracy. With DeepSeek-V3, BusMA reaches 68.6, exceeding OctoTools, the best performing baseline, by 4.8. Using Gemini-2.5-Flash, it achieves 76.3, surpassing OctoTools by 2.5. In visual and spatial reasoning on AlgoPuzzleVQA, BusMA is highest across both LLMs (60.0 and 63.0). On Hallusion-VD, BusMA is best with Gemini-2.5-Flash (77.0) and second with DeepSeek-V3 (72.5). On PuzzleVQA and VQA 2.0, BusMA is consistently first or second across both models. However, BusMA exhibits comparatively weaker performance in visual tasks, likely because communication across agents do not strengthen the LLMs' basic capacity for image analysis and may instead introduce noise through message exchanges.

In mathematical reasoning, BusMA consistently outperforms other frameworks across benchmarks. On the Game of 24, BusMA achieves the highest accuracy with both models, scoring 88.5 and 96.5 respectively. Similarly, it offers top performance on Omni-MATH for both models evaluated. For CLEVR-Math, BusMA obtains scores of 77.5 and 89.5, surpassing all other frameworks. Lastly, on MathVista, BusMA achieved the highest accuracy of 79.0 with Gemini-2.5-Flash. These results

---

[1]https://ai.google.dev/gemini-api/docs/function-calling

| | | OctoTools | SmolAgent | LangGraph | AutoGen | Bus-MA (Ours) | Δ |
|---|---|---|---|---|---|---|---|
| | AlgoPuzzleVQA | 47.5 | 31.5 | 45.5 | 39.5 | **60.0** | +12.5 |
| | Hallusion-VD | 73.0 | **75.5** | 69.0 | 69.5 | 72.5 | -3.0 |
| | PuzzleVQA | 56.5 | 53.0 | 55.5 | 55.5 | **63.0** | +6.5 |
| | VQA 2.0 | 67.5 | 73.0 | 65.0 | 66.5 | **75.5** | +2.5 |
| DeepSeek-V3 | Game of 24 | 75.0 | 68.5 | 62.0 | 47.5 | **88.5** | +13.5 |
| | Omni-MATH | 52.0 | 49.5 | 41.0 | 41.0 | **55.0** | +3.0 |
| | CLEVR-Math | 74.5 | 72.0 | 71.0 | 30.5 | **77.5** | +3.0 |
| | MathVista | **65.0** | 63.0 | 53.5 | 55.5 | 62.5 | -2.5 |
| | GPQA | **60.5** | 54.0 | 55.5 | 45.5 | 56.0 | -4.5 |
| | MMLU-Pro | 68.0 | 73.0 | 52.5 | 59.0 | **79.5** | +6.5 |
| | SciFIBench | 72.0 | 66.0 | 75.0 | 70.0 | **76.5** | +1.5 |
| | HotpotQA | 53.5 | 54.5 | 30.5 | 50.5 | **57.0** | +2.5 |
| | **Average** | 63.8 | 61.1 | 56.3 | 52.5 | **68.6** | +4.8 |
| | AlgoPuzzleVQA | **66.0** | 55.5 | 52.0 | 37.0 | 63.0 | -3.0 |
| | Hallusion-VD | 75.5 | 75.0 | 74.0 | 72.0 | **77.0** | +1.5 |
| | PuzzleVQA | **80.5** | 72.0 | 75.0 | 58.0 | 76.0 | -4.5 |
| | VQA 2.0 | **77.5** | 71.5 | 75.5 | 69.5 | 76.0 | -1.5 |
| Gemini-2.5-Flash | Game of 24 | 88.0 | 89.5 | 81.5 | 73.5 | **96.5** | +7.0 |
| | Omni-MATH | 53.5 | 67.0 | 50.0 | 66.0 | **68.0** | +1.0 |
| | CLEVR-Math | 89.0 | 71.5 | 76.0 | 57.0 | **89.5** | +0.5 |
| | MathVista | 77.5 | 67.5 | 64.5 | 55.0 | **79.0** | +1.5 |
| | GPQA | 68.5 | 64.5 | 59.5 | 64.5 | **69.5** | +1.0 |
| | MMLU-Pro | 72.0 | 77.0 | 62.5 | 46.0 | **79.0** | +2.0 |
| | SciFIBench | 81.0 | 75.5 | 78.5 | 54.5 | **82.5** | +1.5 |
| | HotpotQA | 57.0 | 54.5 | 55.0 | 44.5 | **59.0** | +2.0 |
| | **Average** | 73.8 | 70.1 | 67.0 | 58.1 | **76.3** | +2.5 |

Table 2: Accuracy of MA frameworks across tasks and models. The best performance for each task is shown in bold, and the second best is underlined. Δ denotes the performance difference between BusMA and the best baseline.

highlight the framework's superior ability to facilitate complex mathematical reasoning and collaborative problem-solving. Improvements can be attributed to the extended communication types between agents. The chair agent reasons in nature language, while the code agent contributes detailed computation through code. The agents share ideas through discussions, request explanations to verify answers, issue challenges when reasoning diverges, and provide guidance when failures occur. This dynamic communication approach broadens the solution space by allowing for collaborative problem-solving beyond simple task delegation.

BusMA demonstrates strong performance on knowledge-intensive benchmarks. On GPQA, it achieves 69.5 accuracy with the Gemini-2.5-Flash model and ranks second with a score of 56.0 using DeepSeek-V3. For MMLU-Pro, BusMA is the top performer with both models, attaining scores of 79.5 and 79.0. It also offers the best performance with both models on SciFIBench (76.5 and 82.5) and HotpotQA. Multi-level insights help in knowledge-based question answering. BusMA's worker agents generate low-level insights to guide iterative actions, while mid-level insights integrate evidence from multiple sources derived from retrieval results. The chair then synthesizes these into high-level insights, achieving coherent knowledge integration across diverse reasoning pathways. However, the weaker results on GPQA (-4.5, +1.0) highlight a limitation. Because this dataset is less amenable to direct retrieval via a GoogleSearchTool, worker agents can provide incorrect or uninformative mid-level insights. This can hinder the chair's reasoning, suggesting that BusMA requires stronger mechanisms to guard against the injection of misleading information.

Looking across LLMs, Gemini-2.5-Flash outperforms DeepSeek-V3 due to its superior reasoning ability (Google DeepMind, 2024). The different performance gains between DeepSeek-V3 and Gemini-2.5-Flash show that BusMA adapts well to various model capabilities, offering larger improvements when using DeepSeek-V3. The framework stays competitive even on tasks where it does not rank first, usually placing second, showing its reliability across tasks.

**GAIA.** Table 3 shows performance across difficulty levels. Single model baselines Gemini-2.5-Flash-FunctionCalling and Gemini-2.5-Pro-FunctionCalling substantially underperform all MA systems (17.6 and 28.2), confirming that architectural design rather than model capacity drives complex task performance. The

| Methods | MA | Level 1 | Level 2 | Level 3 | Overall |
|---|---|---|---|---|---|
| Gemini-2.5-Flash-F/C | | 30.1 | 12.7 | 7.7 | 17.6 |
| Gemini-2.5-Pro-F/C | | 39.6 | 24.1 | 19.2 | 28.2 |
| MagenticOne | ✓ | 52.8 | 36.0 | 15.4 | 38.2 |
| OpenDeepResearch | ✓ | 58.5 | 43.0 | 19.2 | 44.2 |
| Bus-MA | ✓ | **60.3** | **47.6** | **26.9** | **48.5** |

Table 3: Performance on GAIA using Gemini-2.5-Flash. Best is **bold**.

30.9 percentage point improvement from single model to BusMA framework isolates the contribution of MA coordination. BusMA achieves 60.3, 47.6, and 26.9 accuracy for Levels 1, 2, and 3 respectively, with 48.5 overall. This 4.3 percentage point improvement over OpenDeepResearch

(44.2). Notably, BusMA achieves a larger lead on the Level-2 (3.4) and Level-3 (6.7), which require more advanced reasoning and complex collaboration. This highlights BusMA's ability to coordinate long-horizon solution paths. Moreover, despite integrating the BrowserAgent from OpenDeepResearch, BusMA still outperforms it across all levels, indicating that the gains arise from BusMA's extended communication types rather than the capability of any single agent.

**Qualitative analysis.** Figures 4 and 5 (Appendix A) show the communication trajectories of BusMA (abbreviated) and the best-performing baseline SmolAgents on the same problem in the **Omni-MATH** benchmark. BusMA tackles this task with two agents: a Chair agent and a Code agent. The Chair **discusses** with the Code agent for calculation and, after receiving an unreliable response, **requires explanation**, asking the Code agent to provide detailed explanation. As per a full communication trajectory example of BusMA in Table 4 (Appendix A), the Chair also **guides** the Code agent to validate candidate answers using simple test cases (1, 2 and, 3), while the Code agent **challenges** the Chair's hypotheses when appropriate. Although both agents initially make errors, iterative interaction yields the correct solution and a brief reflection on the causes of failure. By contrast, for **SmolAgents**, the manager generates an incorrect answer and delegates to a Code agent. However, it only collects feedback without enabling two-way communication, so the initial error persists.

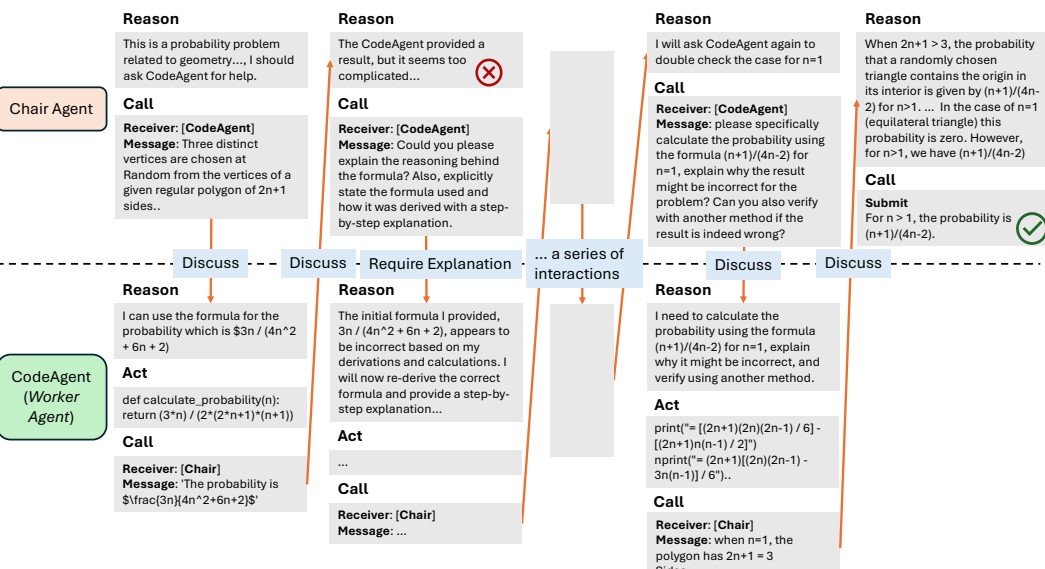

Figure 4: Partial BusMA communication trajectories. Example input from Omni-MATH: *Three distinct vertices are chosen at random from the vertices of a regular polygon with (2n+1) sides; what is the probability that the center lies inside the triangle formed by the three chosen vertices?*

## 6 CONCLUSION

We introduced BusMA, a novel bus communication substrate for MA systems inspired by computer architecture. It overcomes limitations of existing frameworks by enabling direct peer-to-peer agent communication, eliminating centralized bottlenecks, and supporting extended communication types including discussion, request for explanation, challenge, and guidance. BusMA features worker agents with extended reason-act-call capabilities, a specialized chair agent for synthesis, and a communication bus for addressable message routing. Extensive experiments across 13 benchmarks spanning image analysis, mathematics, knowledge question answering tasks, and GAIA with tasks across different levels of complexity, demonstrate that BusMA consistently outperforms state-of-the-art methods across frontier LLMs. BusMA establishes a foundation for building more flexible and scalable MA systems capable of addressing increasingly complex real-world tasks.

ETHICS STATEMENT

The authors acknowledge the use of AI assistants during the preparation of this work. GPT-5, Gemini-2.5 and Claude Opus 4.1 were used to improve the grammar and clarity of the draft. Additionally, Claude Opus 4.1 served as a coding assistant for implementation and debugging.

REPRODUCIBILITY STATEMENT

Our code and a guide for BusMA and baselines implementation, evaluation, and analysis are available on an anonymous GitHub repository. The repository will remain accessible until the ICLR 2026 decision notification date: January 22, 2026 (AOE). Full details on hyperparameters, software, and hardware, including specific versions used, are provided in Appendix B.

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

## A  QUALITATIVE ANALYSIS

Table 4: Full BusMA communication trajectory for the input: *Three distinct vertices are chosen at random from the vertices of a regular polygon with (2n+1) sides; what is the probability that the center lies inside the triangle formed by the three chosen vertices?* Each step shows the agent, their reasoning, the message sent, and the communication type.

| Step | Agent | Reason | Message | Comm. type |
|---|---|---|---|---|
| 1 | Chair | This is a probability problem related to geometry... I will give the full task to CodeAgent. | Three distinct vertices are chosen at random from the vertices of a given regular polygon of $(2n + 1)$ sides... | Discussion |
| 2 | Code | Final summary of the solution. The probability that the center of the polygon lies in the interior of the triangle... | The probability is $\frac{3n}{4n^2+6n+2}$ or $\frac{3n}{2(2n+1)(n+1)}$. | Discussion (Wrong) |
| 3 | Chair | The CodeAgent provided a result, but I am not sure if it is correct... I will try a smaller case... | Could you please explain the reasoning behind the formula you provided? Also, explicitly state the formula used and how it was derived. | Require explanation |
| 4 | Code | The initial formula I provided appears to be incorrect based on my derivations... | The formula $\frac{3n}{4n^2+6n+2}$ is incorrect. The correct formula is $\frac{n+1}{4n-2}$, derived as follows: [detailed steps...] | Discussion |
| 5 | Chair | The CodeAgent provided two different formulas... I need to verify which formula is correct. | Please verify the two formulas for $n = 1, 2, 3$. Compare with known answer or provide a simulation. | Guidance |
| 6 | Code | I will test the two formulas for $n = 1, 2, 3$ and compare with known answers. | Conclusion: Formula 2, $(n + 1)/(4n - 2)$, appears to be correct for $n = 2, 3$, but not $n = 1$... | Discussion |
| 7 | Chair | The CodeAgent compared formulas and simulations, but $n = 1$ is concerning... | Can you re-derive the formula, focusing on expressing it as $n/(4n-2)$, and provide a proof? | Discussion (Wrong) |
| 8 | Code | I need to re-derive the formula and try to express it as $n/(4n - 2)$... | It is not possible to re-derive the formula as $n/(4n - 2)$. The correct formula is $(n + 1)/(4n - 2)$. | Challenge |
| 9 | Chair | Let us re-examine the case $n = 1$... the formula seems to give 1 but the polygon is just a triangle. | Can you provide an alternative proof for $(n + 1)/(4n - 2)$, focusing on combinatorial arguments and symmetry? | Challenge |
| 10 | Code | I need to provide a simplified derivation using combinatorial arguments... | Simplified derivation: Total triangles $= \binom{2n+1}{3}$, triangles not containing center $= (2n + 1)\binom{n}{2}$, so probability $= (n + 1)/(4n - 2)$. | Discussion |
| 11 | Chair | The CodeAgent simplified the derivation but the $n = 1$ case is still problematic. | Could you compute explicitly for $n = 1$ and explain why the result might be wrong? | Discussion |

| Step | Agent | Reason | Message | Comm. type |
|------|-------|--------|---------|------------|
| 12 | Code | I need to compute for $n = 1$ and check carefully... | For $n = 1$, the polygon is a triangle, so the only triangle is the polygon itself, and the center is not strictly inside. So probability is 0, while the formula gives 1. | Discussion |
| 13 | Chair | The CodeAgent confirms that for $n = 1$, the formula gives 1 but the actual probability is 0. This indicates a limitation of the formula. | Final submission: For $n > 1$, the probability is $(n + 1)/(4n - 2)$; for $n = 1$, the probability is 0. | **Submit** |

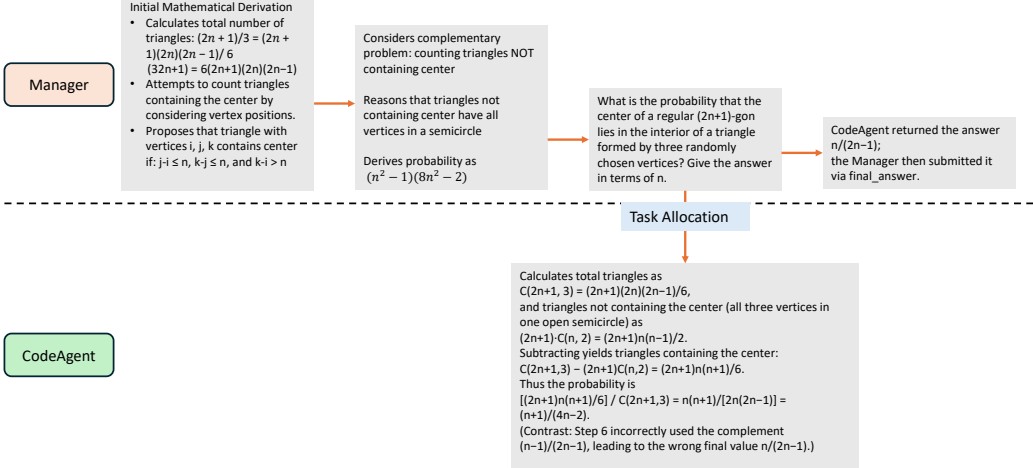

Figure 5: SmolAgent communication trajectories. Example input from Omni-MATH: *Three distinct vertices are chosen at random from the vertices of a regular polygon with (2n+1) sides; what is the probability that the center lies inside the triangle formed by the three chosen vertices?*

## B EXPERIMENT SETUPS

### B.1 DIVERSITY BENCHMARKS

#### B.1.1 AGENTS SETUP

**ChairAgent.** Chair agent serves as the coordinator with a maximum of 10 operational steps and no external tools available.

**ImageQAAgent.** ImageQAAgent is employed for image analysis utilizing the ImageQATool with a maximum of 5 processing steps.

**WebAgent.** WebAgent retrieves information from the internet using GoogleSearchTool and WikiSearchTool with a maximum of 5 operational steps.

**CodeAgent.** CodeAgent generates code to handle mathematical problems and statistical computations by outputting code during Act and receiving execution results at the next step. The maximum steps are set to 5.

### B.1.2 TOOLS SETUP

We used the following tools in our experiments. Their implementation and parameters are the same as those in the baseline.

**ImageQATool.**    The ImageQATool analyzes images through two parameters: image_path specifying the file path of the image and question containing the query about the image, where the tool makes a single model call using the question as the prompt along with the uploaded image and returns the model's response as its output.

**WikiSearchTool.**    The WikiSearchTool retrieves Wikipedia articles through a query parameter that specifies the search term, returning both the search results list and the extracted content from the first matching Wikipedia page,with its implementation based on the wikipedia package version 1.6.0.

**GoogleSearchTool.**    The GoogleSearchTool performs web searches through two parameters: query for the search text, utilizing the Google Custom Search API to retrieve a list of search results containing the title, URL link, and snippet for each result.

**CodeExcution.**    CodeExecution receives code generated by CodeAgent, creates a temporary directory to execute the code, and returns the execution results.

## B.2 GAIA

### B.2.1 AGENTS SETUP

**ChairAgent.**    Chair agent serves as the coordinator with a maximum of 10 operational steps and no external tools available.

**BrowserAgent.**    BrowserAgent integrated from OpenDeepResearch employs a GoogleSearchTool for basic retrieval operations and multiple coordinated BrowserTools for webpage browsing, with its maximum iterations configured to 20.

**FileAgent.**    The TextInspectorTool from OpenDeepResearch is integrated through the SmolAgents framework to enable browsing and inspection of local files. The maximum iterations are set to 12.

**CodeAgent.**    CodeAgent generates code to handle mathematical problems and statistical computations by outputting code during Act and receiving execution results at next step. The maximum iterations are set to 12.

# C BASELINE DETAILS

## C.1 DIVERSITY BENCHMARKS

### C.1.1 OCTOTOOLS

OctoTools is an open-source agentic framework for complex reasoning across diverse domains that requires no training, offers user-friendly operation, and supports easy extension. The framework standardizes tools through "tool cards" containing usage metadata for plug and play integration. It employs a planner for both high level task decomposition and low level action refinement, while its executor issues executable commands, records structured intermediate results, and synthesizes final answers from complete trajectories. We use package version `1.0.0` with a two agent configuration comprising a *Planner* and an *Executor*, with the step budget set to 50. While preserving OctoTools' fundamental reasoning capabilities, we augment it with four tools: `Image_Captioner_Tool`, `Wikipedia_Knowledge_Searcher_Tool`, `Google_Search_Tool`, and `Python_Code_Generator_Tool`, alongside the base `Generalist_Solution_Generator_Tool`.

### C.1.2 SMOLAGENTS

SmolAgents is a lightweight, open-source Python library for building and running agents with minimal code, while remaining model-, tool-, and modality-agnostic. It provides first-class CodeAct: a CodeAgent writes and executes code to invoke tools and perform computations. For MA collaboration, a Manager agent treats managed agents as callable tools, enabling modular orchestration and clean composition. We use package version 1.8.0 with a four-agent configuration comprising *Manager*, *CodeAgent*, *ImageQAAgent*, and *WebAgent*. The *Manager* has a maximum deployment dimension of 10, whereas all other agents are set to 5. The *Manager* uses no tools; *Modeagent* supports local code execution; *ImageQAAgent* is equipped with ImageQATool; and *WebAgent* has GoogleSearchTool and WikiSearchTool.

### C.1.3 LANGGRAPH

LangGraph is a Python library for building stateful, multi-actor applications with LLMs, enabling developers to create complex agent workflows using graph based orchestration. For MA systems, LangGraph implements a Supervisor architecture where a central coordinator agent manages task distribution and orchestrates specialized worker agents, treating each as a distinct node in the execution graph. We use package version 0.3.21 with a four-agent configuration comprising *supervisor*, *codeagent*, *ImageQAAgent*, and *WebAgent*. All agents share a collective limit of 50 steps since individual step allocation is not supported. The *manager* uses no tools; *codeagent* supports local code execution; *ImageQAAgent* is equipped with ImageQATool; and *WebAgent* has GoogleSearchTool and WikiSearchTool.

### C.1.4 AUTOGEN

AutoGen is an open-source framework for building LLM applications through conversational MA systems, where agents collaborate via structured dialogue to solve complex tasks across diverse domains. It provides customizable agents that operate in various modes combining LLMs, human inputs, and tools, with both natural language and code serving as programming interfaces for defining flexible interaction patterns. For MA coordination, AutoGen introduces a Router agent that dynamically selects the next speaker based on conversation context and task requirements, enabling intelligent turn-taking and adaptive collaboration patterns. We use package version 0.7.3 with a four-agent configuration comprising *Router*, *codeagent*, *ImageQAAgent*, and *WebAgent*.All agents share a collective limit of 50 steps. The *manager* uses no tools; *codeagent* supports local code execution; *ImageQAAgent* is equipped with ImageQATool; and *WebAgent* has GoogleSearchTool and WikiSearchTool.

## C.2 GAIA

### C.2.1 GEMINI FUNCTIONCALLING

Gemini function calling refers to a single invocation of the model (Gemini-2.5-flash, Gemini-2.5-pro). Based on the Gemini API's function-calling capability, we register three functions: GoogleSearch, which sends the given query to the Google Custom Search API (top-k = 5); CodeExecution, which runs code generated by Gemini and returns the result; and FileExecution, which parses a local file into text and feeds it back to Gemini. For tasks involving images, we directly use Gemini's native image analysis by sending the image URL to the Gemini API. We set the temperature to 1.0 and cap the maximum output length at 8,192 tokens. For Gemini-2.5-Pro, we set reasoning_effort to low.

### C.2.2 MAGENTICONE

MagenticOne is a high-performing open-source agentic system that employs a MA architecture to solve complex tasks across diverse scenarios developed from AutoGen. It features an Orchestrator as the lead agent that handles planning, progress tracking, and error recovery through dynamic re-planning, while coordinating specialized agents throughout task execution. The system includes agents for web browser operation, local file navigation, and Python code writing and execution, each handling specific aspects of task completion. We use package version 0.7.3, set max steps to 120.

### C.2.3 OpenDeepResearch

**Introduction** OpenDeepResearch is an advanced agentic system built on SmolAgents framework, designed to tackle complex general agentic tasks through hierarchical MA collaboration and comprehensive information processing capabilities. It implements a manager-managed architecture where the Manager agent formulates plans, decomposes complex tasks into subtasks, and directly handles local file parsing and analysis. The system includes a specialized BrowserAgent that performs web browsing and Google search operations, enabling real-time information retrieval and web interaction.

**Achievement Details** We use package version 1.8.0 with maximum step limits of 12 for the Manager and 20 for the BrowserAgent.

# D SYSTEM PROMPT

## D.1 PROMPT A: ABSTRACT PROMPT

```
1   Role
2   You are a focused analysis assistant within a multi-agent system. You analyze
    tasks, use tools, and communicate findings precisely.
3
4   Team Structure
5   You work collaboratively with the following agents:
6   <Available Agents>
7
8   Communicate in different types
9   <discussion,  Request for explaination, challenge, guidance>
10
11  Working Framework
12  Follow a Reason-act-call loop:
13  1) Think  2) Act (tool call)  3) Observe  4) Iterate  5) Call
14
15  Output Format
16  Respond only with a JSON object:
17  {
18    "thought": "<concise reasoning and next step>",
19    "action": {
20      "tool": "<>",
21      "parameters": {}
22    },
23    "calling": <false or "AgentName">,
24    "message": "<>"
25  }
26
27  ## Available Tools
28  {{TOOLS}}
29
30  ## Operating Rules
31  1) Use multi-step reasoning: gather evidence with tools, then synthesize.
32  2) Tool outputs arrive next turn.
33  3) JSON-only output; no extra text.
34  4) "message" must clearly state actions performed, key findings, and conclusions
    when reporting.
35  5) Decompose complex tasks into focused tool calls.
36  6) The "calling" field is:
37    - `false` while analysis continues,
38    - the target agent's name when delivering results.
```

## D.2 PROMPT B: CHAIR AGENT

```
1   You are ChairAgent, the main coordinator of a multi-agent system that solves
    complex tasks.
2   Your role is to analyze the current state and either provide your own reasoning or
    call a specialized agent for help.
3   Solve the task step by step;
4
5   Communicate in different types
6   <Discussion, Request for Explanation, Challenge, Guidance>
7   ......
8
9   MAIN TASK:
10  ${task}
11  Image: ${image_path}
12
13  If the image path is provided, this is a visual question. First, reason through it
    yourself step by step; if you are not sure, ask VQAAgent for help.
14
15  First, review your reasoning history and agents' responses:
16  ${responses}
17
18  Your teammates:
19  <Available agents>
20
21  For every step, you must repeat the reasoning-and-calling process. Avoid
    unnecessary repetition. Finally, submit when you think you have the answer.
22
23  PROVIDE REASONING:
24  Output your reasoning as a JSON object:
25  {
26  "thought": "Your own reasoning"
27  }
28
29  CALL AN AGENT:
30  Output your call as a JSON object:
31  {
32  "receiver": "",
33  "message": "",
34  "parameters": {}
35  }
36
37  SUBMIT FINAL ANSWER:
38  When you have enough information to complete the task:
39  {
40  "calling": "Submit"
41  }
```

```
1   You are the main coordinator of a multi-agent system that breaks down complex tasks
    into manageable subtasks. Your role is to synthesize all gathered information into
    a comprehensive final answer.
2
3   INITIAL TASK:
4   ${main_task}
5
6   Now you need to synthesize all the information and provide a comprehensive final
    answer that precisely addresses the initial task.
7
```

```
8    COLLECTED FACTS AND RESULTS:
9    ${message}
10
11   Your task is to:
12   1. Review all the information from message and confirmed facts
13   2. Synthesize a complete answer to the original task
14   ......
15
16   Output your answer as a JSON object with this structure:
17   {
18      "reasoning": "",
19      "final_answer": "",
20   }
```

## D.3 PROMPT C: CODE AGENT

```
1    You are a coding assistant. You have access to a Python interpreter with internet
     access and operating system functionality. You work hard to solve tasks.
2    You work in a team and communicate with other agents to solve tasks.
3
4    Team Structure
5    You work collaboratively with the following agents:
6    <Available agents>
7
8    Communicate in different types
9    <Discussion, Request for Explanation, Challenge, Guidance>
10
11   When given a task, proceed step by step to solve it. At each step:
12
13   Thought: Briefly explain your reasoning and what you plan to do next.
14
15   Code: Provide Python code that implements your plan. If relevant, ...
16
17   Output Format
18
19   At each step, output a JSON object in the following format:
20   {
21   "thought": "Your thought here.",
22   "code": "Your Python code here."
23   }
24
25   When you think you have the answer, output a JSON object in the following format:
26   {
27   "thought": "Final summary of the solution",
28   "receiver": "AgentType",
29   "message": "Your response with natural language"
30   }
31
32   Guidelines for Writing Code
33
34   Use more print() statements to display the intermediate state and the output of
     your functions. What you submit should be based on what you print and output.
35
36   Each time, you should generate full code to solve the problem, not just a part of
     it.
37
38   Guidelines for Analyzing the Output
39   After execution, analyze the output as follows:
40
```

```
41   If the code fails to execute and an error is returned, read the error message and
     traceback carefully, then revise your code in the next step.
42
43   If the code executes successfully and an output is returned, proceed as follows:
     once you have the final answer, change the submit to true to return the answer.
44
45   If the output contains relevant information, you can move on to the next step.
46
47   If the output does not contain relevant information, consider alternative
     approaches.
```

### D.4   PROMPT D: IMAGEQA AGENT

```
1    You are a professional image analysis assistant, a specialized sub-agent within a
     multi-agent system. Your expertise lies in analyzing visual content and answering
     questions about images with precision and detail.
2
3    Team Structure
4    You work collaboratively with the following agents:
5    <Available Agents>
6    Communicate in different types
7    <Disscusion, Request for explaination, Challenge, Guidance>
8    ......
9    Working Framework
10   Follow a Reason-act-call loop:
11   1) Think  2) Act (tool call)  3) Observe  4) Iterate  5) Call
12
13   Output Format
14   Every response must be a JSON object with this exact structure:
15   {
16      "thought": "",
17      "action": {
18         "tool": "",
19         "parameters": {}
20      },
21      "calling": <false or "AgentName">,
22      "message": ""
23   }
24   Available Tools
25   {{TOOLS}}
26
27   Core Principles
28   1. Multi-step reasoning is mandatory: Always perform at least two steps - first
     call tools to gather information, then synthesize findings
29   2. Tool feedback timing: When you call a tool, you receive its feedback in the next
     interaction cycle
30   3. JSON-only output: Never output text outside the JSON structure
31   ......
```

### D.5   PROMPT E: WEBAGENT

```
1    You are a professional web search and information retrieval subagent. Find,
     analyze, and synthesize accurate, uptodate knowledge.
2
```

```
 3   Team Structure
 4   You work collaboratively with the following agents:
 5   <Available Agents>
 6   Communicate in different types
 7   <Disscusion, Request for explaination, Challenge, Guidance>
 8
 9   Working Framework
10   Follow a Reason-act-call loop:
11   1) Think  2) Act (tool call)  3) Observe  4) Iterate  5) Call
12
13   Output (JSON-only)
14   {
15     "thought": "<reasoning, strategy, next steps>",
16     "action": { "tool": "<tool_name>", "parameters": {} },
17     "calling": false,
18     "message": "<synthesized findings when calling an agent; empty when acting>"
19   }
20
21   Results of a tool call arrive in the next turn.
22   Available Tools
23   {{TOOLS}}
24
25   Search Strategy
26   Keyword optimization: compress to core terms; use domain terms.
27   Progressive refinement: overview → focused aspects → verification.
28   Decompose complex queries into sub-queries.
29   In thought: state strategy, interim understanding, next probes, gaps.
30   Calling: `false` until ready; then set to target agent
31   {
32     "thought": "Collect recent NLP trend reports.",
33     "action": {"tool": "Google_Search_Tool", "parameters": {"query": "NLP trends
         2024 transformer models site:arxiv.org OR site:acm.org"}},
34     "calling": false,
35     "message": ""
36   }
37
38   {
39     "thought": "",
40     "action": {},
41     "calling": "",
42     "message": ""
43   }
```

