# OpenReview forum: "BusMA: A Bus Communication Substrate for Multi- Agent Systems"
_ICLR.cc/2026/Conference — ICLR 2026 Conference Withdrawn Submission_

### Official Review · Reviewer_K9yP · 2025-10-30

**Soundness:** 2
**Presentation:** 3
**Contribution:** 2
**Rating:** 4
**Confidence:** 4

**Summary:**

The paper proposes a multi-agent framework BusMA inspired by the bus communication in computer systems. The bus communication substrate is devoted to solving the centralized bottleneck challenge present in current multi-agent architectures and frameworks, enabling peer-to-peer goal-specific communication. The proposed approach is evaluated on 13 benchmarks from visual and text modalities, demonstrating consistent improvements over the baselines.

**Strengths:**

1. The description of the proposed multi-agent system is clear and easy to follow.
2.  The experimental setup is diverse in terms of the benchmarks and the backbone LLMs.

**Weaknesses:**

1. The paper lacks any ablation experiments to validate the design choices and understand the contribution of individual components.
2. The bus module concept is not novel in multi-agent LLM frameworks field. For example, the AutoGen framework has a SocietyOfMind agent that runs a group chat as an internal monologue and serves as the only agent connected to the chat manager, similarly to the bus module in the proposed architecture.
3. The paper acknowledges weaker performance on GPQA, attributing it to "incorrect or uninformative mid-level insights". However, no failure analysis is provided for the proposed bus communication.

**Questions:**

1. When and why bus communication fails to improve reasoning?
2. What are possible ways to mitigate error propagation in the proposed bus communication?
3. What are the trade-offs between communication overhead and reasoning quality?

---

### Official Review · Reviewer_QRLQ · 2025-10-31

**Soundness:** 3
**Presentation:** 3
**Contribution:** 2
**Rating:** 2
**Confidence:** 4

**Summary:**

This paper introduces BusMA, a multi-agent communication framework inspired by computer bus architectures. It aims to overcome the limitations of existing Hierarchical Manager-Worker (HMW) and Router-based Message Passing (RMP) systems by enabling direct, addressable peer-to-peer communication. The system consists of Worker Agent that can call specific peers with varied communication types (e.g., discussion, challenge), and a Chair Agent that synthesizes the conversation history from the Communication Bus to produce a final solution. The authors claim novelty in the communication protocol and a multi-level insight generation process. Through extensive experiments on 13 benchmarks, BusMA is shown to outperform several state-of-the-art multi-agent frameworks, particularly in complex reasoning tasks.

**Strengths:**

- Well-motivated by addressing communication limitations in existing HMW and RMP multi-agent systems by workers' reason-act-call.
- Solid empirical evaluation with comprehensive benchmarks, strong baselines, and consistent state-of-the-art results.
- The work is reproducible with the provided open-source code.

**Weaknesses:**

- The novelty and contribution of this paper are limited. While the central metaphor of a "bus communication system" from computer architecture is an intriguing framing, its implementation within the proposed framework appears simplistic: worker agents engage in peer-to-peer communication through a communication bus, and a central chair agent synthesizes the final output. Moreover, it doesn't fully escape the centralized MA design it aims to critique. The heavy reliance on the "chair agent" for both initial task distribution to worker agents and final information synthesis firmly establishes it as a central coordinator.
- The second claimed contribution, the multi-level insight generation (low, mid, high), seems limited. This hierarchy appears to be more of a descriptive framework that relabels inherent reasoning processes within a multi-agent workflow, rather than a substantive new mechanism. Specifically, the insights from low to high only correspond to the reasoning of different agents and actions, whose contribution to the framework's overall performance feels overstated.
- The experimental validation for the framework's superiority could be strengthened. While BusMA outperforms the selected baselines, its overall GAIA score of 48.5 remains modest compared to other state-of-the-art works such as Youtu-Agent, AWorld, etc. Moreover, this paper lacks ablation studies to isolate the source of performance gains. Including such experiments would substantially bolster the paper's claims regarding its novel design contributions.

**Questions:**

Please refer to the Weaknesses

---

### Official Review · Reviewer_HXvG · 2025-10-31

**Soundness:** 3
**Presentation:** 3
**Contribution:** 3
**Rating:** 2
**Confidence:** 4

**Summary:**

This paper proposes BusMA, a communication-centered multi-agent framework inspired by bus architectures in computer systems. Unlike traditional multi-agent systems where agents communicate via ad hoc message passing or centralized memory, BusMA introduces a shared communication bus that serves as an explicit reasoning and coordination medium. Each agent connects to the bus through standardized interfaces, allowing dynamic registration, message broadcasting, and state synchronization. The authors argue that this bus-based structure enables scalable coordination, modular reasoning, and improved transparency compared with prior message-based frameworks such as AutoGen and AgentVerse. Experiments on multi-tool reasoning show BusMA outperforming other frameworks in success rate and reasoning efficiency.

**Strengths:**

1. The analogy between computer bus systems and multi-agent coordination is intuitive and provides a coherent design metaphor.
2. The framework is modular and extendable, offering a practical foundation for future multi-agent systems.

**Weaknesses:**

1. The “bus” mechanism does not fundamentally change the communication topology—it is effectively a centralized message dispatcher. Therefore, BusMA inherits the same scalability and single-point failure issues as prior centralized coordination frameworks (e.g., AutoGen, AgentVerse), without providing new coordination semantics or efficiency guarantees.
2. The framework lacks a formal communication protocol or optimization objective. Message scheduling and content selection are fully prompt-driven, without any learnable or rule-based coordination mechanism. This limits BusMA’s theoretical grounding and scalability to complex multi-agent interactions.
3. The framework assumes a shared global state but does not specify consistency guarantees or synchronization mechanisms. In concurrent update scenarios, BusMA may suffer from non-deterministic state overwriting, undermining the reproducibility of multi-agent interactions.
4. The interaction quality and final task success are highly sensitive to the design of individual agent prompts. However, the paper neither unifies nor ablates prompt templates across baselines, making it unclear whether BusMA’s gains originate from the architecture or improved prompt engineering.
5. Although the authors claim that the bus improves scalability, no empirical or analytical study is provided on communication latency or bandwidth usage. In fact, the bus architecture introduces a centralized broadcast mechanism with O(N) message complexity, which may limit scalability when the number of agents increases.
6. The bus is treated as a passive communication layer rather than an active reasoning component. Without any aggregation, attention, or reasoning objective tied to bus interactions, it remains unclear how the bus itself contributes to reasoning quality beyond facilitating message passing.

**Questions:**

-

---

### Official Review · Reviewer_WEMw · 2025-11-01

**Soundness:** 3
**Presentation:** 3
**Contribution:** 2
**Rating:** 4
**Confidence:** 4

**Summary:**

This paper introduces BusMA, a novel multi-agent communication framework inspired by the "bus" concept in computer architecture. The framework aims to overcome the communication flexibility and error propagation limitations of existing Hierarchical Management (HMW) and Routing-Based Management (RMP) methods. Core features of BusMA include a decentralized communication bus for direct point-to-point agent interaction (e.g., discussions, challenges, requests for explanations) and a three-tiered "insight" generation mechanism coordinated by a Chair agent. The authors validated BusMA on 12 benchmarks across various domains, including image processing, mathematics, and question answering, as well as on the GAIA benchmark, demonstrating its superiority over several existing methods.

**Strengths:**

* The paper clearly articulates the bottlenecks of current mainstream multi-agent communication paradigms (HMW, RMP), specifically the communication limitations and error propagation stemming from centralized control.
* The paper's narrative and diagrams are of high quality. The distinct roles of the three components of the BusMA framework (Chair, Workers, and Bus) are well-defined, rendering the entire workflow easily comprehensible.
* The proposed method was tested on 12 datasets spanning multiple domains, including vision, mathematics, and question answering, in addition to the GAIA benchmark.
* By presenting specific communication trajectories and comparing them with baseline methods, the paper effectively highlights BusMA's advantages in fostering iterative and critical discussions among agents.

**Weaknesses:**

1) Limited Novelty: This is a primary concern. While the "bus" metaphor is engaging, the implementation appears more akin to a shared memory or a message queue that supports addressable communication. The framework does not seem to incorporate more fundamental mechanisms of bus architecture, such as arbitration, congestion control, or priority scheduling. Furthermore, the supported point-to-point communication types (e.g., discussions, challenges) could, in theory, be emulated by enhancing a single agent's reflection or chain-of-thought capabilities. Consequently, this work feels more like a sophisticated engineering implementation with a compelling narrative rather than a fundamental conceptual breakthrough.
2) Potential for Unfair Experimental Comparisons:
- The experiments were conducted using only two models (DeepSeek-V3, Gemini-2.5-Flash), which raises questions about the generalizability of the findings to other models, particularly those of varying sizes.
- I have noted that in other publicly available reports (e.g., the Tongyi DeepResearch Technical Report), some baseline models (such as OpendeepResearch) have achieved significantly higher scores on the GAIA benchmark than those reported in this paper. This discrepancy raises concerns about the specific details and reproducibility of the experimental setup.
3) Superficial Analysis: The paper effectively demonstrates that BusMA performs well but falls short of explaining why. The absence of necessary ablation studies makes it difficult to ascertain the individual contributions of each component, such as the three-tiered insight mechanism and the four distinct communication types. It remains unclear whether the performance gains stem from more efficient collaboration or simply from the increased number of LLM calls and implicit opportunities for reflection that communication entails.
4) Overlooking Core Limitations: In the discussion section, the paper acknowledges the risk that "Workers may provide incorrect information, thus misleading the Chair." However, this is a critical vulnerability in such communication systems. The authors have not proposed any mechanisms to mitigate this issue, such as confidence scoring, voting, or information verification. This omission significantly undermines the robustness of the framework.

**Questions:**

1) Innovation: Beyond address-based routing, in what other ways does BusMA genuinely borrow from and implement the core principles of traditional bus architectures (e.g., arbitration, congestion control)? How would you counter the argument that "simulating critical discussions among multiple agents could be achieved by enhancing the reflective capabilities of a single agent," and what are the fundamental advantages of BusMA in comparison to such an approach?
2) Experimental Fairness: Could you provide a more equitable experimental setup where all methods are compared under the same total number of LLM calls or inference tokens? This would help to demonstrate that the performance improvements are a direct result of communication efficiency rather than a computational advantage. How do you account for the significant discrepancies between the baseline results reported in your paper and those available in other public reports?
3) Analytical Depth: Do you plan to conduct ablation experiments to quantify the specific contributions of the different communication types and the three-level insight mechanism to the final performance? In your view, what is the single most critical factor contributing to BusMA's success?
4) Robustness: Have you considered or designed any specific mechanisms to defend against the risk of misinformation propagation that you identified? If not, does this imply that BusMA has inherent vulnerabilities when dealing with tasks characterized by high uncertainty?

---

### Note · Authors · 2025-12-27

I have read and agree with the venue's withdrawal policy on behalf of myself and my co-authors.